# Rabies seropositive individuals, dogs, and healthcare professionals without prior vaccination in four Brazilian Indigenous communities

**Matheus Lopes Ribeiro**[1], **Camila Michele Appolinario**[1], **Bruna Letícia Devidé Ribeiro**[1], **João Henrique Farinhas**[2], **Fernando Rodrigo Doline**[2], **Gisely Toledo Barone**[3], **Juliana Amorim Conselheiro**[3], **Vamilton Alvarés Santarém**[4], **Leandro Meneguelli Biondo**[5], **Andrea Pires dos Santos**[6], **Rogério Giuffrida**[4], **Louise Bach Kmetiuk**[6,7], **Alexander Welker Biondo**[6,8], **Jane Megid**[1]*

**1** Department of Animal Production and Preventive Veterinary Medicine, São Paulo State University (UNESP), Botucatu, São Paulo, Brazil, **2** Graduate College of Cell and Molecular Biology, Federal University of Paraná (UFPR), Curitiba, Paraná, Brazil, **3** Laboratory of Diagnostics of Zoonosis and Vector-borne Diseases (LabZoo), Zoonosis Surveillance Division, Health Surveillance Coordination, Municipal Health Department, São Paulo, São Paulo, Brazil, **4** University of Western São Paulo (UNOESTE), Presidente Prudente, São Paulo, Brazil, **5** Interdisciplinary Graduate Studies, University of British Columbia, Okanagan, Kelowna, British Columbia, Canada, **6** Purdue University, West Lafayette, Indiana, United States of America, **7** Zoonosis Surveillance Unit, City Secretary of Health, Curitiba, Paraná, Brazil, **8** Department of Veterinary Medicine, Federal University of Paraná (UFPR), Curitiba, Paraná, Brazil

* jane.megid@unesp.br

## Abstract

Indigenous communities are reportedly among the most vulnerable populations exposed to rabies worldwide. Accordingly, this study aimed to assess rabies serum titers from healthy Indigenous individuals, their dogs, and healthcare professionals of four Indigenous communities from São Paulo state, southeastern Brazil. Blood samples were collected, and an epidemiological questionnaire applied. The samples were processed by Fluorescent Antibody Virus Neutralization (FAVN) method. Overall, 35/299 (11.7%) individuals and 22/166 (13.2%) dogs without prior vaccination were seropositive. Furthermore, 4/18 (16.7%) healthcare professionals were seropositive, with only one reporting prior rabies vaccination. The lack of rabies titers in the remaining 14/18 (77.8%) healthcare professionals indicates no immune protection. Seropositivity was associated with being from the Kopenoty community (p = 0.026) and with owners reporting seeing their dogs in contact with bats (p = 0.022). In summary, these results should be considered as a warning for the risk of human-dog rabies exposure and infection, mainly due to bat contact.

## Author summary

Rabies is a zoonotic infectious disease considered to be of great public health importance due to its high lethality and numerous epidemiological transmission cycles. Indigenous communities may present several risk factors for rabies exposure. These include poor

**Data availability statement:** The authors confirm that all relevant data appear in the paper and its Supporting Information files, without restriction.

**Funding:** MLR received funds from Fundação de Amparo à Pesquisa do Estado de São Paulo—FAPESP (grant number 2022/00834-8) and by the Coordenação de Aperfeiçoamento de Pessoal de Nível Superior—Brasil (CAPES) —Finance Code 001. The funders had no role in study design, data collection and analysis, decision to publish, or preparation of the manuscript.

**Competing interests:** The authors have declared that no competing interests exist.

living conditions in natural areas, limited access to health services and a lack of awareness about prevention practices. The present study has shown, for the first time, the presence of rabies-neutralizing antibodies in Indigenous human and dog populations without prior vaccination and living in an Indigenous land in the countryside of São Paulo state, southeastern Brazil. The Epidemiological Surveillance Center has reported rabies-positive bats in 2017 in the study area, near a highly endemic state region that registered 7/40 (17.5%) positive bats. Such overlapping scenarios have likely favored the presence of rabies virus-neutralizing antibodies in individuals and their dogs living in natural Indigenous areas without prior rabies vaccination. Results herein should be carefully considered as a warning for human-dog rabies exposure and infection risk, particularly due to potential bat contact, in Indigenous communities and healthcare professionals in Brazil and worldwide.

## Introduction

Rabies is a zoonotic infectious disease caused by a Lyssavirus of the Rhabdoviridae family. It is considered to be of great public health importance due to its high lethality and numerous epidemiological transmission cycles, which occur in various environments [1]. Infected mammals, including pets, farm animals, and humans, may present severe clinical signs, such as convulsions, paralysis, and ultimately death [2]. In such a scenario, Indigenous communities may present several risk factors for rabies exposure. These include poor living conditions in natural areas, limited access to health services and a lack of awareness about prevention practices [3]. In addition, close human-animal contact in Indigenous communities, including wildlife, livestock, and companion animals, can often facilitate rabies virus transmission [4].

The epidemiology of rabies in Brazil has recently been described as a complex interaction between natural and social factors, according to the Ministry of Agriculture and Livestock [5]. Addressing this public health challenge has required comprehensive interventions, such as large-scale animal vaccination programs, community education, promotion of human vaccination, prevention of animal bites, and prompt access to post-exposure treatment, recommended by the Ministry of Health [6]. Additionally, the recently created Ministry of Indigenous Peoples has advocated for culturally sensitive strategies adapted to Indigenous communities as a crucial pathway for successful interventions [7].

Despite the vaccination efforts and the current surveillance system, rabies has remained a significant public health concern in Brazil, particularly in Indigenous communities [8,9]. In addition, no comprehensive study has been conducted on concomitant serosurvey of humans and their dogs in such vulnerable populations. Accordingly, the present study has aimed to assess humans and dogs of four Indigenous communities in Bauru city, countryside of São Paulo state, southeastern Brazil.

## Materials and methods

### Ethics statement

The present study was approved by the National Human Ethics Research Committee (protocol 52039021.9.0000.0102) and the Ethics Committee of Animal Use (protocol 033/2021), both through the Federal University of Paraná, southern Brazil. Written formal consent was obtained for human participants, and when children participated, the consent was obtained from the parent or guardian. In addition, the study herein has been approved by the Special Indigenous Health District—South Seashore and included as part of the official activities.

## Study area

The study was conducted in four different Indigenous communities of Guarani and Terena ethnicities, both belonging to the Araribá Indigenous Land with 19.21 km², located within the Bauru municipality, countryside of São Paulo state (Table 1). The Bauru municipality (22° 18′ 54″ S 49° 03′ 39″ W) has an area of 667.7 km² and at the time was nationally ranked 66th (top 1.2%) in terms of population with 364,500 inhabitants, 37th (top 0.7%) in Human Development Index (HDI) with a score of 0.801 (very high), and 74th (top 1.3%) in Gross Domestic Product (GDP) out of 5,568 Brazilian municipalities. The municipality presented 526 m of elevation, tropical of altitude (Aw) Köppen climate, temperature varying from 15 °C to 30 °C (average 22.6 °C), and rainfall from 28 mm to 208 mm (annual average 1330 mm).

## Sample collection

Participants were informed about the study, invited to participate voluntarily, asked to sign a consent form, and, if they agreed, completed an epidemiological questionnaire. All participants signed the consent form. Human blood samples were collected by cephalic puncture by a certified nurse. After the owner's consent, dog blood samples were collected by jugular puncture by an accredited veterinarian, and the verbal consent was obtained from the owners when it existed. All blood samples were placed into individual clotting tubes, refrigerated, and transported to the nearby (140 km; 87 miles) São Paulo State University at Botucatu campus and stored at −80 °C until serological testing in the reference Laboratory of Diagnostics of Zoonoses and Vector-borne Diseases (LabZoo) São Paulo, where they were processed for anti-rabies antibodies.

## Serological analyses

Serum neutralization of all samples was performed by Fluorescent Antibody Virus Neutralization (FAVN), a technique adapted on Rapid Fluorescent Focus Inhibition Test (RFFIT) using four serial dilutions (1:3) of serum samples, along with positive, negative and standard control sera in microplates, with conversion from dilution to IU/mL carried out using the Sperman-Karber method, as previously established [10,11]. Titers equal to or higher than 0.5 IU/mL were considered adequate for dogs and all mammals [10,12]. Additionally, titers were statistically assessed starting at 0.2 IU/mL, considered for wildlife exposure, as an epidemiological assessment of contact between both Indigenous human and dog populations to the rabies virus [13,14].

## Epidemiological Data and Statistical Analysis

Two epidemiological questionnaires were administered to individuals of the Indigenous communities to assess associated risk factors. The questionnaires were designed with open and closed questions addressing personal information (age, gender, and ethnicity) and potential

**Table 1. Name, coordinates, total and sampled population, and ethnicities of studied Indigenous communities belonging to the Araribá Indigenous Land, located in São Paulo state.**

| Indigenous communities | Coordinates | Total Population | Sampled | % | Ethnicity |
|---|---|---|---|---|---|
| Ekeruá | 22°16′28.05″ S 49°22′21.95″ O | 159 | 56 | 35.2 | Guarani, Terena |
| Kopenoty | 22°15′58.20″ S 49°21′00.95″O | 245 | 125 | 51.0 | Guarani, Kaingang, Terena |
| Nimuendajú | 22°17′30.81″ S 49°22′41.03″O | 100 | 73 | 73.0 | Guarani, Terena |
| Tereguá | 22°15′55.31″ S 49°20′53.34″ O | 127 | 47 | 37.0 | Terena, Guarani |

associated risk factors for RABV seropositivity (S1 and S2 Figs). The questionnaires were completed through an interview conducted by a trained healthcare professional, with the assistance of an Indigenous interpreter when necessary. As mentioned, a voluntarily signed consent term was obtained before the questionnaires and blood samplings, with all procedures performed in compliance with the National Human Ethics Research Committee for the use of human data and testing results.

The database consistency was checked for missing or duplicated data. Continuous variables (age in years) were categorized into groups and divided into quartiles. A univariate analysis was employed to test the association between risk factors and the outcomes of rabies adequate levels by neutralizing antibodies (titers equal to or above 0.5 IU/mL) and a lower level of rabies-neutralizing antibodies, indicative of RABV contact (titers above 0.2 and below 0.5 IU/mL), applying Pearson's chi-square test or Fisher's exact test, separately for Indigenous individuals and their dogs.

Continuity correction was utilized for categories with zero counts to stabilize the estimative, adding 0.5 to each cell. Variables showing statistical significance <0.20 in the univariate analysis were assessed in a multivariate logistic regression model in the backward stepwise modality, with bias correction using the penalized maximum likelihood method, considered more suitable for rare events and sparse predictors [15]. The association between the predictor variables and the serological results was presented as odds ratios with respective 95% confidence intervals. The assumption of no multicollinearity was validated by calculating the variance inflation factor (VIF). All the analyses were carried out in the R program with the brglm2 package [16,17], considering a 5% significance level.

## Results

Overall, 35/299 (11.7%) Indigenous individuals and 32/166 (19.3%) dogs without prior notice of vaccination were seropositive, indicating rabies virus contact. In addition, 1/18 (11.1%) healthcare professionals presented adequate titer (1.12 UI/mL), and 2/18 (11.1%) had contact titers (0.22 and 0.38 UI/mL), with another one self-reporting rabies vaccination (10.21 UI/mL) (Fig 1). Seropositivity was associated with being from the Kopenoty community (p = 0.026) for Indigenous individuals and, notably, with owners reporting seeing their dogs in contact with bats (p = 0.022).

CC BY 4.0 license—Map created by the authors, using the following data:

1. State and Municipality Boundaries

- Data Source: Public Continuous Cartographic Bases from IBGE—Brazil

- Data Source Link: IBGE Continuous Cartographic Bases—https://www.ibge.gov.br/en/geosciences/maps/continuous-cartographic-bases/18067-continuous-cartographic-bases-brazil.html?lang=en-GB&t=sobre; Continuous Cartographic Bases—IBGE—https://www.ibge.gov.br/en/geosciences/maps/continuous-cartographic-bases/18067-continuous-cartographic-bases-brazil.html?lang=en-GB&t=sobre

- License: Public Open Data

- Access Date: 2024/01/17

- Copyright and Technical Information: IBGE Institutional Information—https://www.ibge.gov.br/en/access-to-information/institutional/the-ibge.html

2. Indigenous Land and Community Locations

- Data Source: FUNAI Geoserver

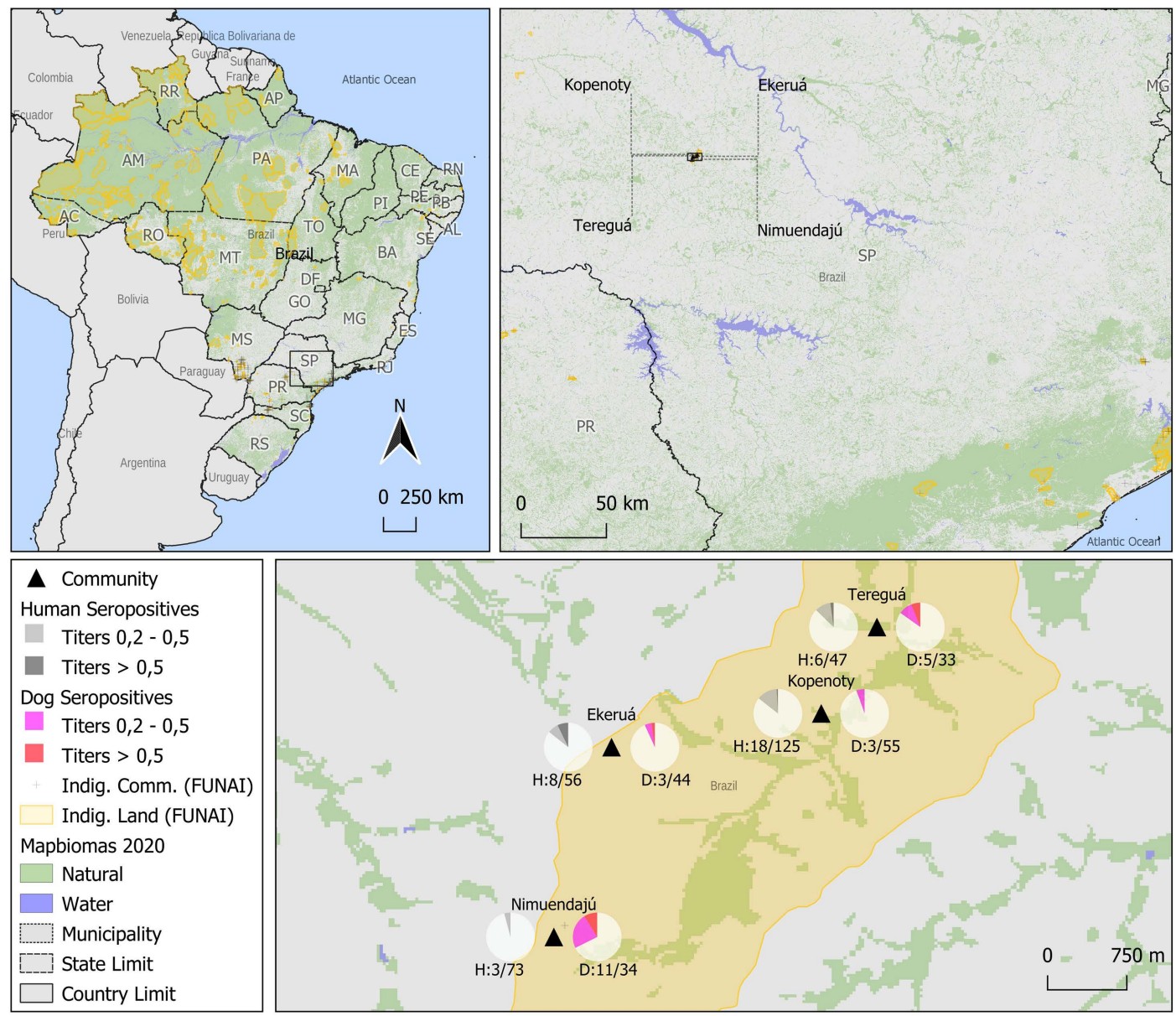

**Fig 1. Seropositive individuals, dogs, and healthcare professionals without prior vaccination in four Brazilian Indigenous communities.**

- Data Source Link: FUNAI Geoserver Map Preview—https://geoserver.funai.gov.br/geoserver/web/wicket/bookmarkable/org.geoserver.web.demo.MapPreviewPage?1&filter=false

- License: Public Open Data

- Access Date: 2024/01/17

- Copyright Information: Unrestricted Public Data—FUNAI—https://www.gov.br/funai/pt-br/acesso-a-informacao/informacoes-classificadas

3. Mapbiomas 2020 Land Cover (Water and Vegetation)

- Data Source: MapBiomas Project—Collection 7 of the Annual Land Use Land Cover Maps of Brazil

- Data Source Link: https://storage.googleapis.com/mapbiomas-public/initiatives/brasil/collection_7/lclu/coverage/brasil_coverage_2020.tif

- License: CC BY

- Access Date: 2024/01/17

- Copyright and Technical Information: MapBiomas FAQ—How to Cite—https://brasil.mapbiomas.org/en/faq/como-faco-para-citar-os-dados-do-mapbiomas/

The prevalence of anti-rabies antibodies was calculated for the total number of individuals without previous vaccination according to the epidemiological questionnaire answered by each Indigenous person (Tables 2 and 3), separated into two distinct groups: above 0.2 IU/mL and below 0.5 IU/mL and equal to or above 0.5 IU/mL [14]. All the Indigenous people who had a prevalence of neutralizing antibodies against rabies underwent a documentary analysis of their vaccination cards by an Indigenous health professional to prove that they had not previously been vaccinated against rabies and none of them presented any supporting documentation for this immunization. Above the adequate titer, four Indigenous people were found with 1.95 IU/mL (one resident of the Kopenoty community and three others from the Ekeruá community) and two individuals with 5.87 IU/mL (one from the Ekeruá community and the other from the Tereguá community). Concerning the canine samples, six animals had antibody titers above 0.5 IU/mL, all with different values. As for their distribution, one dog was from Ekeruá, two from Tereguá, and three from the Nimuendajú community.

**Table 2. Prevalence of seroneutralizing antibodies against rabies, determined by the FAVN (Fluorescent Antibody Virus Neutralization) technique, in Indigenous populations of the Araribá Land, without previous vaccination records.**

| Community | Total | Titers (≥ 0.5 IU/mL) | | | Titers (> 0.2 < 0.5 IU/mL) | | |
|---|---|---|---|---|---|---|---|
| | | Seropositives | % (95% CI) | Range (min–max) | Seropositives | % (95% CI) | Range (min–max) |
| Ekeruá | 56 | 4 | 7.14 (2.81–17.0) | 1.94–5.87 | 4 | 7.14 (2.81–16.97) | 0.21 |
| Kopenoty | 125 | 1 | 0.80 (0.14–4.4) | 1.94 | 17 | 13.6 (8.67–20.7) | 0.21–0.37 |
| Nimuendajú | 71 | 0 | NA | NA | 3 | 4.2 (1.45–11.70) | 0.21 |
| Tereguá | 47 | 1 | 2.12 (0.40–11.11) | 5.87 | 5 | 10.63(4.63–22.60) | 0.21–0.37 |
| Total | 299 | 6 | 2.0 (0.92–4.30) | 1.94–5.87 | 29 | 9.7 (6.80–13.60) | 0.21–0.37 |

NA, not applicable.

**Table 3. Prevalence of seroneutralizing antibodies against rabies, determined by the FAVN (Fluorescent Antibody Virus Neutralization) technique, in dogs from Indigenous communities of Araribá Land, without previous vaccination records.**

| Community | Total | Titers (≥ 0.5 IU/mL) | | | Titers (> 0.2 < 0.5 IU/mL) | | |
|---|---|---|---|---|---|---|---|
| | | Seropositives | % (95% CI) | Range (min–max) | Seropositives | % (95% CI) | Range (min–max) |
| Ekeruá | 44 | 1 | 2.27 (0.40–11.81) | 1.12 | 2 | 4.55 (1.26–15.14) | 0.21–0.22 |
| Kopenoty | 55 | 0 | NA | NA | 3 | 5.45 (1.87–14.85) | 0.21–0.38 |
| Nimuendajú | 34 | 3 | 8.82 (3.05–22.96) | 1.95–5.87 | 8 | 23.53 (12.44–40.00) | 0.21–0.37 |
| Tereguá | 33 | 2 | 6.06 (1.68–19.61) | 0.66–1.15 | 3 | 9.09 (3.14–23.57) | 0.22–0.38 |
| Total | 166 | 6 | 3.61 (1.67–7.66) | 0.66–5.87 | 16 | 9.64 (6.02–15.08) | 0.21–0.38 |

NA, not applicable.

Variables associated with rabies adequate levels of neutralizing antibodies (bivariate analysis for the outcome titer ≥ 0.5 UI/mL) were "community" (p = 0.026) and "have you ever seen your dog in contact with bats" (p = 0.022). In addition to these statistically significant associated risk factors, the variable "ever had contact with a bat" was included in the final multivariate logistic regression model for the same outcome. The final model was obtained after one step, with only the variable "Have you ever seen your dog in contact with bats" being retained (p = 0.0159). This information has been summarized and presented (Table 4).

Regarding indicative factors of contact for Indigenous people, no statistically significant variable was observed in the bivariate analysis for the outcome "rabies neutralizing antibodies titers above 0.2 and below 0.5 IU/mL" (Table 5). However, in the multivariate logistic model, the variables "community" (p = 0.171) and "works at school or student" (p = 0.090) were tested, as both met the selection parameters (p < 0.20). The final model was obtained in one step, with only the variable "works at school or student" retained in the final model, which was considered non-significant (p = 0.070).

After serological analysis, all individuals who showed soroneutralizing antibodies against rabies, regardless of titer, were sought out to answer a second questionnaire. 23 of 35 Indigenous people were found and agreed to take part in this second stage (13 from Kopenoty, 4 from Tereguá, 5 from Ekeruá and 2 from Nimuendajú). All participants (100%) denied knowing what zoonosis is. When asked which animals transmit rabies, 6/23 (26%) denied knowing. Of the remainder who said they did know, only 13% cited bats as the animal that transmits rabies. 95% of them cited the dog as responsible for transmission and no participant cited any other animal, domestic or wild, involved in this process.

In addition, 21/23 91.3% said they had seen bats near their homes. The most frequently mentioned places were the roofs and ceilings of their own houses, and all the participants reported that they still use these places even with the constant presence of chiroptera. 34.7% answered that they had seen a dead bat in the village more than once. When asked what they did when they saw this scene, 75% said they picked up the animal with their hands and threw it in the nearest garbage can, while the other 25% said they didn't do anything.

When asked about the presence of other wild animals in Indigenous communities, 16/23 (69.5%) said they had already seen them. The most commonly mentioned animals were wild dogs (8), foxes (7), snakes (7), capybaras (6), jaguars (5), armadillos (4), ocelots (3), monkeys (3), skunks (2), anteaters (1) and coati (1). 13/23 (56.5%) reported having seen domestic animals in contact with wild animals and 17/23 (73.9%) answered that the dogs in the village eat the carcasses of other animals. When asked if wild animal bites or scratches on domestic animals could mean a risk of disease infection for them, 14/23 (60.8%) said yes, 3/23 (13%) said no and 6/23 (26%) said they didn't know.

Regarding the consumption of gamemeat by the Indigenous people themselves, 18/23 (78.2%) said that they eat it or have eaten it. Among the animals most consumed by the population were capybara (11), armadillo (9), paca (5), lizard (3), nambu (2), tapir (1), coati (1) and monkey (1).

When asked whether or not they knew of any animal deaths suspected of rabies in the village, 5/23 (21.7%) said they did. All 5 were from the Kopenoty community and described the same case: in 2022 a horse died after being attacked by a bat. The Special Indigenous Health District—South Seashore said it had no knowledge of what had happened. The participants who mentioned this event reported that the animal was not taken for official diagnosis and was buried on the outskirts of the Indigenous community. In addition, two participants reported the death of a dog as a suspected case of rabies. According to both, the animal died with neurological signs after an accident with a bat. 2/23 (8.6%) said they knew of a case of human death by rabies in the villages of Araribá Land.

**Table 4. Variables associated with the presence of rabies seroneutralizing antibodies, determined by the FAVN technique, in titers considered to be adequate (equal to or above 0.5 IU/mL), in the Indigenous populations of Araribá Land by bivariate and multivariate logistic analysis.**

| | Titer ≥ 0.5 IU/mL | | Univariate analysis | | Multivariate analysis | |
|---|---|---|---|---|---|---|
| | Positive (%) | Negative (%) | OR (CI 95%) | p-value | OR (CI 95%) | p-value |
| Variables | | | | | | |
| | N = 6 (2) | N = 293 (98) | | | | |
| Community*: | | | | 0.026 | | |
| Ekeruá | 4 (66.7) | 52 (17.7) | Ref. | | | |
| Kopenoty | 1 (16.7) | 124 (42.3) | 7.11 (1.09–46.42) | | | |
| Nimuendajú | 0 (0.00) | 71 (24.2) | 12.26 (0.64–232.64) | | | |
| Tereguá | 1 (16.7) | 46 (15.7) | 2.66 (0.40–17.58) | | | |
| Gender: | | | | 0.416 | | |
| Female | 2 (33.3) | 161 (54.9) | Ref. | | | |
| Male | 4 (66.7) | 132 (45.1) | 0.43 (0.05–2.35) | | | |
| Extract age*: | | | | 0.559 | | |
| <17 years | 3 (50.0) | 85 (29.1) | Ref. | | | |
| 17–25 years | 0 (0.00) | 57 (19.5) | 4.70 (0.24–92.87) | | | |
| 26–43 years | 1 (16.7) | 82 (28.1) | 2.25(0.32–15.61) | | | |
| >43 years | 2 (33.3) | 68 (23.3) | 1.12 (0.21–5.86) | | | |
| Wok at school or student: | | | | 0.215 | | |
| No | 1 (16.7) | 145 (49.5) | Ref. | | | |
| Yes | 5 (83.3) | 148 (50.5) | 0.23 (0.01–1.50) | | | |
| Have you ever seen your dog in contact with bats: | | | | 0.022 | | |
| No | 4 (66.7) | 61 (20.8) | Ref. | | | |
| Yes | 2 (33.3) | 232 (79.2) | 7.28 (1.31–60.0) | | 6.65 (1.42–31.03) | 0.016 |
| Have you ever seen your cat in contact with bats? | | | | 0.406 | | |
| No | 5 (83.3) | 171 (58.4) | Ref. | | | |
| Yes | 1 (16.7) | 122 (41.6) | 3.20 (0.48–85.5) | | | |
| Location of the pets*: | | | | 0.231 | | |
| Don't have pets | 3 (50.0) | 46 (15.7) | Ref. | | | |
| Communal | 0 (0.00) | 26 (8.87) | 3.98 (0.20–80.23) | | | |
| Peridomicile | 3 (50.0) | 193 (65.9) | 4.16 (0.91–18.94) | | | |
| Domicile | 0 (0.00) | 28 (9.56) | 4.29 (0.21–86.14) | | | |
| You've had contact with bats: | | | | 0.094 | | |
| No | 5 (83.3) | 130 (44.4) | Ref. | | | |
| Yes | 1 (16.7) | 163 (55.6) | 5.62 (0.85–150) | | 4.37 (0.73–26.10) | 0.105 |
| You've seen bats near the community: | | | | 1000 | | |
| No | 3 (50.0) | 168 (57.3) | Ref. | | | |
| Yes | 3 (50.0) | 125 (42.7) | 0.74 (0.13–4.40) | | | |

Ref, reference category for the odds ratio calculation.

*Continuity correction applied.

No associations were observed between the sex and age of dogs and the presence of rabies-neutralizing antibodies. Results of the statistical analysis for associated risk factors with dog antibody titers considered as adequate (≥ 0.5 IU/mL) and those titers associated

**Table 5. Variables associated with the presence of seroneutralizing antibodies against rabies, determined by the FAVN (Fluorescent Antibody Virus Neutralization) technique, in titers above 0.2 and below 0.5 IU/mL, considered indicative of contact in the Indigenous populations of Araribá Land by bivariate and multivariate logistic analyses.**

| | Titers > 0.2 < 0.5 IU/mL | | Univariate analysis | | Multivariate analysis | |
|---|---|---|---|---|---|---|
| | Positive (%) | Negative (%) | OR (CI 95%) | p-valor | OR (CI 95%) | p-valor |
| Variables | | | | | | |
| | N = 29 (10,74) | N = 270 (89,26) | | | | |
| Community: | | | | 0.171 | | |
| Ekeruá | 4 (13.8) | 52 (19.3) | Ref. | | | |
| Kopenoty | 17 (58.6) | 108 (40.0) | 0.50 (0.14–1.46) | | | |
| Nimuendajú | 3 (10.3) | 68 (25.2) | 1.72 (0.34–9.66) | | | |
| Tereguá | 5 (17.2) | 42 (15.6) | 0.65 (0.15–2.72) | | | |
| Sex: | | | | 0.786 | | |
| Female | 17 (58.6) | 146 (54.1) | Ref. | | | |
| Male | 12 (41.4) | 124 (45.9) | 1.20 (0.55–2.68) | | | |
| Age group*: | | | | 0.251 | | |
| <17 years | 6 (20.7) | 82 (30.5) | Ref. | | | |
| 17–25 years | 3 (10.3) | 54 (20.1) | 1.28 (0.31–6.67) | | | |
| 26–43 years | 11 (37.9) | 72 (26.8) | 0.49 (0.16–1.36) | | | |
| >43 years | 9 (31.0) | 61 (22.7) | 0.50 (0.16–1.49) | | | |
| Work at school or student | | | | 0.090 | | |
| No | 19 (65.5) | 127 (47.0) | Ref. | | | |
| Yes | 10 (34.5) | 143 (53.0) | 2.12 (0.96–4.95) | | 2.08 (0.94–4.57) | 0.070 |
| Have you ever seen your dog in contact with bats: | | | | 1000 | | |
| No | 6 (20.7) | 59 (21.9) | Ref. | | | |
| Yes | 23 (79.3) | 211 (78.1) | 0.95 (0.33–2.32) | | | |
| Have you ever seen your cat in contact with bats: | | | | 1000 | | |
| No | 17 (58.6) | 159 (58.9) | Ref. | | | |
| Yes | 12 (41.4) | 111 (41.1) | 0.99 (0.45–2.21) | | | |
| Location of the pets*: | | | | 0.640 | | |
| Don't have pets | 5 (17.2) | 44 (16.3) | Ref. | | | |
| Communal | 1 (3.45) | 25 (9.26) | 2.55 (0.36–70.1) | | | |
| Peridomicile | 19 (65.5) | 177 (65.6) | 1.08 (0.34–2.89) | | | |
| Domicile | 4 (13.8) | 24 (8.89) | 0.68 (0.16–3.11) | | | |
| You've had contact with bats: | | | | 0.531 | | |
| No | 17 (58.6) | 154 (57.0) | Ref. | | | |
| Yes | 12 (41.4) | 116 (43.0) | 1.06 (0.49–2.38) | | | |

Ref, reference category for the odds ratio calculation.

*Continuity correction applied.

with RABV exposure (above 0.2 and below 0.5 IU/mL) have been summarized and presented (Tables 6 and 7).

Serological tests were also performed on 18 community healthcare professionals who work between 15 and 40 hours a week in the Indigenous villages, resulting in titers ranging from 0.07 to 10.21 IU/mL. 2/18 (11.1%) healthcare professionals had adequate serological titers (1.12 and 10.21 UI/mL), and 2/18 (11.1%) had titers indicative of contact (0.22 and 0.38 UI/mL), with only 1/18 self-reporting prior rabies vaccination (5.6%). Of all healthcare professionals, 12/18 (85.7%) reported previous contact with bats. Due to small sampling and subject to errors, statistics were not carried out in this group.

**Table 6. Variables associated with the presence of seroneutralizing antibodies against rabies, determined by the FAVN (Fluorescent Antibody Virus Neutralization) technique, in titers considered to be adequate (equal to or above 0.5 IU/mL) in Indigenous dog populations of Araribá Land by bivariate and multivariate logistic analyses.**

| | Titers ≥ 0.5 IU/mL | | Univariate analysis | |
|---|---|---|---|---|
| | Positive (%) | Negative (%) | OR (CI 95%) | p-value |
| Variables | 6 (6.6) | 160 (93.4) | | |
| Sex: | | | | 1.000 |
| Female | 3 (50.0%) | 73 (45.6%) | Ref. | |
| Male | 3 (50.0%) | 87 (54.4%) | 1.19 (0.20–7.12) | |
| Age (in years): | | | | 0.691 |
| 1 | 1 (16.7%) | 39 (24.4%) | Ref. | |
| 2 | 1 (16.7%) | 45 (28.1%) | 1.15 (0.03–45.9) | |
| 3 | 1 (16.7%) | 38 (23.8%) | 0.97 (0.02–39.0) | |
| >4 | 3 (50.0%) | 38 (23.8%) | 0.36 (0.01–3.22) | |

Ref, reference category for calculating the odds ratio.

**Table 7. Variables associated with the presence of seroneutralizing antibodies against rabies, determined by the FAVN (Fluorescent Antibody Virus Neutralization) technique, in titers considered to be indicative of contact (above 0.2 and below 0.5 IU/mL) in the dog populations of Araribá Land, by bivariate and multivariate logistic analyses.**

| | Titers > 0.2 < 0.5 IU/mL | | Univariate analysis | |
|---|---|---|---|---|
| | Positive (%) | Negative (%) | OR (CI 95%) | p-value |
| Variables | 16 (9.6) | 150 (90.4) | | |
| Sex: | | | | 0.663 |
| Female | 3 (50.0) | 73 (45.6) | Ref. | |
| Male | 3 (50.0) | 87 (54.4) | 1.19 (0.20–7.12) | |
| Age (in years): | | | | 0.815 |
| 1 | 1 (16.7) | 39 (24.4) | Ref. | |
| 2 | 1 (16.7) | 45 (28.1) | 1.15 (0.03–45.9) | |
| 3 | 1 (16.7) | 38 (23.8) | 0.97 (0.02–39.0) | |
| >4 | 3 (50.0) | 38 (23.8) | 0.36 (0.01–3.22) | |

Ref, reference category for calculating the odds ratio.

Noteworthy, all four Indigenous communities studied herein were located close to each other and the Indigenous people and their pets have contact with each other.

## Discussion

The present study shows, for the first time, the presence of rabies-neutralizing antibodies in human and dog populations without prior vaccination from four Indigenous communities in the countryside of São Paulo state, southeastern Brazil. The Bauru Epidemiological Surveillance Group reported 57/4,281 (1.3%) rabies-positive bats in 2017 [18], near a highly endemic state region that registered 7/40 (17.5%) positive bats [19]. Such overlapping scenarios have likely favored the presence of rabies virus-neutralizing antibodies in individuals and their dogs living in natural Indigenous areas without prior rabies vaccination.

The individuals who tested positive for rabies antibodies have provided evidence of exposure to the virus, as recently established [20]. This contact may have been due to direct

or indirect contact with infected bats in the region, as no human or dog rabies outbreak was reported in the area during the period. Although not officially reported, the possibility of rabies in horses and dogs probably originating from bats, as reported by seropositive Indigenous people, cannot be ruled out. Furthermore, the Zoonosis Control Center of São Paulo City diagnosed rabies in 5,670 bats from 1998 to 2003. It was found that 5.9% of healthy bats had rabies virus-neutralizing antibodies above 0.5 UI/ml, confirming the endemic rabies status of the aerial cycle statewide [21].

Of the Indigenous individuals who presented rabies virus-neutralizing antibody titers, 19/35 (54.2%) had contact with bats, with 5/19 (26.3%) individuals under 17 years old. This exposure may indicate that children come into contact with bats out of curiosity and for recreational purposes, which can lead to handling, scratching, and biting, as observed in other Indigenous populations, factors associated with the four Indigenous human rabies deaths in Minas Gerais state, in Brazil [22]. Such behavior should be strongly discouraged [23]. Similarly, an epidemiological survey found that all 50 elementary schools in a southern Brazilian seacoast area reported bat cohabitation, sightings of bats in their households, and bat bites on their animals. As alarming, students described daytime handling and playing with bats at school and in their surroundings [24]. Although it was possible to find older seropositive participants with possible multiple low-dose exposures to RABV, the participants do not live in isolated communities, with the majority having occupations in cities nearby their communities.

The high number of bat encounters mentioned herein (Table 4) may corroborate the continuous spreading of bat-related rabies outbreaks in human and animal populations of Latin America, particularly by the hematophagous bat *Desmodus rotundus* [25]. Other Brazilian bat species, both hematophagous and non-hematophagous, have also caused outbreaks and posed a continuous risk to human and animal health [26]. Recently, non-hematophagous bats have been considered important RABV reservoirs, with the *Myotis nigricans* and *Eptesicus furinalis* bat species presenting the highest prevalence of RABV positivity in the São Paulo state [27]. In another Indigenous community in Brazil, chiropteran attacks are frequent, since this population is used to living with domesticated dogs and cats and taming wild animals such as monkeys and bats [23].

In addition to hematophagous and non-hematophagous bats, other wildlife species close to the Indigenous communities herein may be harboring RABV, such as non-human primates. A serosurvey of free-ranging capuchin monkeys (*Cebus apella nigritus*) in a nearby environmentally protected area of São Paulo state has shown 4/36 (11.1%) seropositive individuals [28]. In addition, the marmoset *Callithrix jacchus* species has also been recognized as a rabies reservoir in Brazil, with the potential for other wildlife animals to emerge as new countrywide reservoirs [15,29]. Finally, hunting of exotic invasive species may be a source of infection, as a recent study in southern Brazil has shown 9/80 (11.0%) free-range wild boars with titers for rabies exposure (≥ 0.10 IU/mL) while 43/49 (88.0%) of corresponding hunters lacked immune adequate titers (<0.50 IU/mL) [30].

The seroneutralization techniques (RFFIT and FAVN) have been considered the main approved methods by the World Health Organization, which has adopted a very strict human cut-off, precisely covering possible inherent test interferences and considering the disease severity [10]. In wild animals and other species, studies have considered less conservative cut-offs and indicated that values greater than 0.10 IU/ml may indicate RABV virus circulation [14]. However, the possibility of non-specific inactivation cannot be ruled out.

The occurrence of rabies-specific antibodies in healthy, unvaccinated individuals may be the result of an alternative course of rabies virus exposure [20]. Initially, Fekadu identified four alternative courses of rabies infection that could lead to detecting rabies antibodies in

healthy individuals [31]. Neutralizing antibodies to the rabies virus also have been discovered in two communities within a bat-overlapping environment in the Peruvian Amazon, with 7/63 (11.0%) seropositive persons, with titer between 0.1 and 2.8 IU/mL, and it is important to note that only one seropositive respondent reported having received post-exposure prophylaxis for rabies, also it encountered the presence of immunoglobulin G against rabies virus ribonucleoprotein in three individuals [32]. However, a systematic review of re-exposure rabies prophylaxis has shown that Indigenous communities have been more susceptible to accidents involving wildlife mammals [33].

Dogs without previous rabies vaccination also presented rabies seropositivity, with 16/166 (9.64%) dogs with titers above 0.2 IU/mL and below 0.5 IU/mL and 6/166 (3.61%) with titers above 0.5 IU/mL. Such a surprising outcome may have likely resulted from the hunting of infected wildlife, as well as eating carcasses of already dead rabid animals. Regardless, the presence of infected dogs may present a high infection risk for Indigenous communities and residents living in the surroundings of the Indigenous reservation. In northern Australia, free-roaming domestic dogs in Indigenous communities were considered the likely rabies spreaders on Australian territory in the event of a possible incursion of RABV coming from rabies-endemic Southeast Asia [34]. Thus, as in the study herein, unrestricted circulation of unvaccinated dogs may predispose rabies spreading from wildlife to dog to human populations.

When a host encounters a pathogen, antibodies that can specifically bind to the antigens on that pathogen are selected and increased in number. If the host survives such exposure, the enhanced immune response may be detected after the pathogen elimination. Consequently, the presence of rabies-specific antibodies in healthy, unvaccinated individuals may indicate prior exposure to RABV with no fatal outcome [20]. The presence of anti-rabies antibodies in unvaccinated individuals herein suggests prior viral exposure, but not necessarily active viral replication, as even a single dose of an inactivated rabies vaccine can induce anti-rabies responses [35]. Although the natural development of rabies antibodies in healthy humans has been rarely documented, several studies have reported detectable rabies viral-neutralizing antibodies in the serum from individuals living in highly endemic areas of rabies [20].

More than 75% of the Indigenous people who presented soroneutralizing antibodies against rabies stated that they had eaten game meat, including birds. Handling carcasses or eating meat was reported in Hanoi/Vietnam [36] and two cases of human rabies were reported as a result of handling dog and cat carcasses probably as a result of the habit of eating dog and cat [37]. In India, a natural rabies infection in a domestic fowl (*Gallus domesticus*) was found for the first time and so the risk of exposure through the consumption of infected meat, although unlikely, could potentially represent a risk of rabies transmission to humans [38]. In the Amazon region of Peru, none of the respondents reported preparing or consuming bats as a food source, but other game meats—which can also be reservoirs of the rabies virus—were not elucidated [32].

Establishing accurate and contextual testing cut-offs for different species and populations may play a crucial role in interpreting the results of antibodies against rabies virus [39]. Defining cut-off points in non-vaccinated populations may be an additional challenge in serological rabies tests, as the absence of prior immunization may lead to significant variability in individual immune response, mostly due to factors such as historical exposure to the virus, presence of natural antibodies and genetic diversity of exposed populations, minimizing false positives [14]. The potential cross-reactivity among closely related viruses may also be a factor that makes interpreting rabies serology tests more complex, as their antigen similarity may enable antibody production against one virus to neutralize others. Since no other circulating lyssavirus has been reported in Brazil, rabies virus neutralizing antibodies may strongly suggest

either a current infection or previous exposure to rabies virus [20], but several other factors can also affect the sensitivity for detecting previous exposure to RABV [40], making a reevaluation of serological tests necessary before applying to non-vaccinated individuals. A successful prophylactic approach against rabies mostly relies on holistic observation and action in different populations within a given community [41]. Nonetheless, the results obtained herein have confirmed the importance of urgent vaccination of Indigenous human and dog populations and further studies to establish the rabies cycle in such communities fully.

The individual immune response to natural rabies virus infection includes the production of virus-specific neutralizing and binding antibodies, which are influenced by factors such as the viral dose, the extent of replication in the periphery and successful entry and replication in the central nervous system [32]. The seropositive responses without prophylaxis indicate exposure to RABV. So the presence of neutralizing antibodies in unvaccinated subjects may indicate prior exposure to the virus, even if viral replication does not necessarily occur [35].

The Indigenous Health Secretariat has informed that the studied communities solely relied on the Secretariat for rabies prophylaxis protocol, with no independent pre- or post- exposure vaccination, confirming that vaccination cards reflected true vaccination status of all analyzed individuals [7]. In addition, the São Paulo state suspended vaccination campaigns for dogs and cats since 2021 [42]. Finally, as people living in Indigenous communities may not fully understand the rabies risks, they are unlikely to seek medical assistance following suspicious bat contact, handling or bite [43].

As major limitations, access and monitoring of Indigenous communities may be impaired due to remote location, barrier language, cultural differences, and misinterpretation of questionnaires leading to vague or wrong answers. In addition, dogs of Indigenous communities were free-range, never leashed, and had no individual owners. Thus, no questionnaire could be applied to properly assess owner, age, origin, habits, and other epidemiological aspects to pinpoint cross-owner-dog-associated risk factors.

## Conclusion

The present study has shown, for the first time, the presence of rabies-neutralizing antibodies in Indigenous human and dog populations without prior vaccination and living in an Indigenous land in the countryside of São Paulo state, southeastern Brazil. Results herein should be carefully considered as a warning for human-dog rabies exposure and infection risk, particularly due to potential bat contact, in Indigenous communities and healthcare professionals in Brazil and worldwide.

## Supporting information

**S1 Fig. Epidemiological questionnaire for identification and general knowledge of all study participants.**
(PDF)

**S2 Fig. Epidemiological questionnaire to identify risk factors for participants with soro-neutralizing antibodies against rabies.**
(PDF)

## Acknowledgments

The authors are deeply thankful to healthcare and administration professionals at the Special District for Indigenous Health—Seashore South, Special Secretariat for Indigenous Health, Ministry of Health, Brazil.

## Author contributions

**Conceptualization:** Matheus Lopes Ribeiro, Alexander Welker Biondo, Jane Megid.

**Data curation:** Matheus Lopes Ribeiro, Camila Michele Appolinario, João Henrique Farinhas, Fernando Rodrigo Doline, Alexander Welker Biondo, Jane Megid.

**Formal analysis:** Matheus Lopes Ribeiro, Bruna Letícia Devidé Ribeiro, Gisely Toledo Barone, Juliana Amorim Conselheiro, Leandro Meneguelli Biondo.

**Investigation:** Matheus Lopes Ribeiro, Camila Michele Appolinario, João Henrique Farinhas, Fernando Rodrigo Doline, Rogério Giuffrida, Louise Bach Kmetiuk, Alexander Welker Biondo.

**Methodology:** Matheus Lopes Ribeiro, Camila Michele Appolinario, Louise Bach Kmetiuk, Alexander Welker Biondo, Jane Megid.

**Project administration:** Alexander Welker Biondo, Jane Megid.

**Resources:** Matheus Lopes Ribeiro, Andrea Pires dos Santos, Alexander Welker Biondo, Jane Megid.

**Supervision:** Camila Michele Appolinario, Alexander Welker Biondo, Jane Megid.

**Writing – original draft:** Matheus Lopes Ribeiro.

**Writing – review & editing:** Matheus Lopes Ribeiro, Camila Michele Appolinario, Gisely Toledo Barone, Juliana Amorim Conselheiro, Vamilton Alvarés Santarém, Andrea Pires dos Santos, Rogério Giuffrida, Louise Bach Kmetiuk, Alexander Welker Biondo, Jane Megid.

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
