## [Decision Letter · Decision Letter 0]

16 Aug 2024

Dear Professor Megid,

Thank you very much for submitting your manuscript "One Health approach to Rabies: seropositive individuals, dogs, and healthcare professionals without prior vaccination in Brazilian Indigenous communities" for consideration at PLOS Neglected Tropical Diseases. As with all papers reviewed by the journal, your manuscript was reviewed by members of the editorial board and by several independent reviewers. In light of the reviews (below this email), we would like to invite the resubmission of a significantly-revised version that takes into account the reviewers' comments. 

Please note the reviewer's concern regarding definitive evidence of lack of vaccination and adequate testing to determine rabies exposure. This is critical, and unless provided, the stated conclusions of this manuscript cannot be supported. The findings must be accurate for publication given that these are potentially a paradigm challenging findings.

We cannot make any decision about publication until we have seen the revised manuscript and your response to the reviewers' comments. Your revised manuscript is also likely to be sent to reviewers for further evaluation.

Sincerely,

Victoria J. Brookes

Section Editor

Victoria Brookes

Section Editor

Reviewer's Responses to Questions

**Key Review Criteria Required for Acceptance?**

**Methods**

-Are the objectives of the study clearly articulated with a clear testable hypothesis stated?

-Is the study design appropriate to address the stated objectives?

-Is the population clearly described and appropriate for the hypothesis being tested?

-Is the sample size sufficient to ensure adequate power to address the hypothesis being tested?

-Were correct statistical analysis used to support conclusions?

-Are there concerns about ethical or regulatory requirements being met?

Reviewer #1: The study objectives are clearly articulated with a clear testable hypothesis stated.

The study design is appropriate to address the stated objectives.

The population is clearly described and appropriate for the hypothesis being tested.

The sample size is sufficient to ensure adequate power to address the hypothesis being tested.

The statistical analyzes used were correct to support the conclusions.

In my opinion there are no concerns about compliance with ethical or regulatory requirements.

Reviewer #2: (No Response)

**Results**

-Does the analysis presented match the analysis plan?

-Are the results clearly and completely presented?

-Are the figures (Tables, Images) of sufficient quality for clarity?

Reviewer #1: The analysis presented corresponds to the proposed analysis plan.

The results are presented clearly and completely in the corresponding section.

Figure 1 (image) is inserted in an appropriate place in the text (results section), has the necessary captions for its interpretation and understanding and appears to have a good quality of readability. As for the tables, both have captions and are of sufficient quality for greater clarity of the results presented there.

Reviewer #2: Some results and tables need to be update for correct cut point of RVNA

**Conclusions**

-Are the conclusions supported by the data presented?

-Are the limitations of analysis clearly described?

-Do the authors discuss how these data can be helpful to advance our understanding of the topic under study?

-Is public health relevance addressed?

Reviewer #1: I believe the conclusions are adequately supported by the data presented.

The limitations of the analysis are clearly described in the last paragraph of the discussion.

The authors discuss how such data can be useful in advancing our understanding of the topic under study.

The relevance to public health is addressed in the introduction and reinforces this in the discussion.

Reviewer #2: Conclusion need revision with better evidence

**Editorial and Data Presentation Modifications?**

Reviewer #1: Accept

Reviewer #2: The use of the word "protective" refering to level of RVNA is incorrect, please delete all instances were that word was used, and use the correct expression: "adequate". 

Also the adequate antibody level recommended by WHO is ≥0.5 UL/ml, in the manuscript is implied only >0.5 UL/ml, please correct, meaning table need to be redone also.

**Summary and General Comments**

Reviewer #1: This article is interesting because it seeks to evaluate the anti-rabies serum titres of healthy indigenous people, their dogs and health professionals from four indigenous communities in the state of São Paulo, southeastern Brazil. It is very important to research the presence of antibodies against the rabies virus in indigenous populations and in domestic animals not previously vaccinated against rabies, given the close contact of Brazilian natives with hematophagous or non-hematophagous chiropterans. The study design is adequate and the work is technically sound for the proposed objective. Statistical analysis and its interpretation are appropriate to the results obtained. I believe that all initial data underlying the results are available to ensure full reproducibility and can support future studies. The conclusions are adequately supported and well represented by the results section. The work is well presented in a clear and precise way and presents current bibliographic references. Therefore, I suggest publishing this article as it stands.

Reviewer #2: The received manuscript is highly relevant and important for rabies science. 

Unfortunately, the findings of rabies neutralizing antibodies (RVNA) in unvaccinated humans and dogs, even in healthcare workers are not properly supported by the data presented in the manuscript.

The claims in the article are very significant, and all efforts to demonstrate the accuracy of the statements and results need to be exhausted. The evidence about negative previous vaccination is lacking or insuficient, access to rabies vaccine, types of vaccines and years of use for humans and dog in the areas studied need to be explained in detail. If the individual with RVNA does not have a registry of his inmunization in a medical record, that does not rule out previous vaccination. Without a detailed explanation of vaccine delivery system, or the possibility for rabies vaccine access in other ways (such as: a campaign , outbreak reponse or due to work in a factory or industry that requires rabies immunisation and that is provided separately than the offical MoH services) we cannot conclude for sure the invididual was never vaccinated against rabies. A good discussion of those issues , biases and limitations is needed in the discussion section.

Another important issue is that , previous findings of rabies antibodies in local population (Gilbert et al 1992) did not find RVNA but rabies antibodies that were not neutralizing and levels of RVNA were not adecuate, therefore previous research does not support the findings in the current manuscript, then the presented findings would be really important and will dispute previous immuniological profiles published. To compare with previous results properly, the authors need to show additional immunological test than only FVNA, of course additional differences on exposures and environmental factors in the population studied could lead to explain the findins, but that specific issues are not welll described in the manuscritp that even leads to canine exposures , and also is undefined regarding the type of bat exposures mentiones, and Brazil have serveral cycles for rabes in bats, and of course exposure factors are different, all that need to be well undesrtood in the manuscript. 

Please discuss memory bias, and other potential issues that can lead to accept a dog or an indivudual as previously non vaccinated against rabies when was not. Same with healthcafre personnel.

I will be glad to review the changes and improvements for the manuscript, as I said, results are very significant given are well supported by evidence, and I sincerely hope the authors take the task and effort to provide what is needed to achieve publication.

PLOS authors have the option to publish the peer review history of their article (what does this mean? ). If published, this will include your full peer review and any attached files.

**Do you want your identity to be public for this peer review?** For information about this choice, including consent withdrawal, please see our Privacy Policy .

Reviewer #1: No

Reviewer #2: No
---

## [Decision Letter · Decision Letter 1]

5 Dec 2024

PNTD-D-24-00608R1

Rabies seropositive individuals, dogs, and healthcare professionals without prior vaccination in four Brazilian Indigenous communities

Dear Dr. Megid,

Thank you for submitting your manuscript to PLOS Neglected Tropical Diseases. After careful consideration, we feel that it has merit but does not fully meet PLOS Neglected Tropical Diseases's publication criteria as it currently stands. Therefore, we invite you to submit a revised version of the manuscript that addresses the points raised during the review process.

Please submit your revised manuscript within 60 days Jan 04 2025 11:59PM. If you will need more time than this to complete your revisions, please reply to this message or contact the journal office at plosntds@plos.org. Please include the following items when submitting your revised manuscript:

We look forward to receiving your revised manuscript.

Kind regards,

Victoria J. Brookes

Section Editor

Shaden Kamhawi

co-Editor-in-Chief

Paul Brindley

co-Editor-in-Chief

**Additional Editor Comments:**

Please ensure that the reviewer's comments are addressed. As said previously, this manuscript is potentially paradigm changing, and conclusions must be thoroughly supported by the results and/or all limitations thoroughly discussed.

**Journal Requirements:**

1) We note that your "rabies_indigenous_map_v01plos-_1_.tiff" file is duplicated on your submission. Please remove any unnecessary or old files from your revision, and make sure that only those relevant to the current version of the manuscript are included.

**Reviewers' Comments:**

Reviewer's Responses to Questions

**Key Review Criteria Required for Acceptance?**

**Methods**

-Are the objectives of the study clearly articulated with a clear testable hypothesis stated?

-Is the study design appropriate to address the stated objectives?

-Is the population clearly described and appropriate for the hypothesis being tested?

-Is the sample size sufficient to ensure adequate power to address the hypothesis being tested?

-Were correct statistical analysis used to support conclusions?

-Are there concerns about ethical or regulatory requirements being met?

Reviewer #2: (No Response)

**Results**

-Does the analysis presented match the analysis plan?

-Are the results clearly and completely presented?

-Are the figures (Tables, Images) of sufficient quality for clarity?

Reviewer #2: (No Response)

**Conclusions**

-Are the conclusions supported by the data presented?

-Are the limitations of analysis clearly described?

-Do the authors discuss how these data can be helpful to advance our understanding of the topic under study?

-Is public health relevance addressed?

Reviewer #2: (No Response)

**Editorial and Data Presentation Modifications?**

Reviewer #2: (No Response)

**Summary and General Comments**

Reviewer #2: Thank you for modification in the revised version.

The most important issue from previous review remains unresolved. There was a mistake in the previous review , it was meant to say Gilbert et al 2012 for the reference.

Please read carefully "Evidence of rabies virus exposure among humans in the Peruvian Amazon", Gilbert et. al Am J Trop Med Hyg. 2012 Aug;87(2):206-215. doi: 10.4269/ajtmh.2012.11-0689. availalable online at https://www.ajtmh.org/view/journals/tpmd/87/2/article-p206.xml , that is cited in the manuscript with number 32

It is neccesary the authors examine the results of the Gilbert et al paper, and compare the inmunological profile of the individuals found with rabies antibodies. A thoughful discussion on this is expected, given that paper is the most important reference for the results of the manuscript.

The previous review comments regarding individuals found to have ≥0.5 IU/mL , are not resolved without a proper discussion of manuscript results considering Gilbert et al paper.

Minor comment

Please correct mistakes in the tables 2 and up. Intervals for the RVNA in tables show 2 columns , one ≥0.5 IU/mL, and other 0.2-0.5 IU/mL , then the value of 0.5 is included in both groups. Since 0.5 value is already included in the group ≥0.5 IU/mL, the other group should be <0.5 IU/mL , it is not relevant an interval 0.2-<0.5, saying <0.5 UI/mL is enough.

I look forward to see an updated review of the manuscript.

PLOS authors have the option to publish the peer review history of their article (what does this mean? ). If published, this will include your full peer review and any attached files.

**Do you want your identity to be public for this peer review?** For information about this choice, including consent withdrawal, please see our Privacy Policy .

Reviewer #2: No

**Figure resubmission:**
---

## [Editor Report · Decision Letter 2]

17 Jan 2025

Dear Professor Megid,

We are pleased to inform you that your manuscript 'Rabies seropositive individuals, dogs, and healthcare professionals without prior vaccination in four Brazilian Indigenous communities' has been provisionally accepted for publication in PLOS Neglected Tropical Diseases.

Best regards,

Husain Poonawala

Academic Editor

Victoria Brookes

Section Editor

Shaden Kamhawi

co-Editor-in-Chief

Paul Brindley

co-Editor-in-Chief

When submitting the final version for proofing, could you please double check the use of univariate and multivariate - I recommend using univariable and multivariable since only a single outcome is being measured, as opposed to multiple outcomes.

---

## [Editor Report · Acceptance letter]

Dear Professor Megid,

We are delighted to inform you that your manuscript, "Rabies seropositive individuals, dogs, and healthcare professionals without prior vaccination in four Brazilian Indigenous communities," has been formally accepted for publication in PLOS Neglected Tropical Diseases.

Best regards,

Shaden Kamhawi

co-Editor-in-Chief

Paul Brindley

co-Editor-in-Chief
